# Sagittal Integral Morphotype of Female Classical Ballet Dancers and Predictors of Sciatica and Low Back Pain

**DOI:** 10.3390/ijerph18095039

**Published:** 2021-05-10

**Authors:** Antonio Cejudo, Sebastián Gómez-Lozano, Pilar Sainz de Baranda, Alfonso Vargas-Macías, Fernando Santonja-Medina

**Affiliations:** 1Department of Physical Activity and Sport, Faculty of Sport Sciences, Regional Campus of International Excellence “Campus Mare Nostrum”, University of Murcia, C.P., 30100 Murcia, Spain; psainzdebaranda@um.es; 2Department of Sport and Physical Activity, Faculty of Sport, Jerónimos Campus, Catholic University of Murcia UCAM, C.P., 30107 Murcia, Spain; 3Telethusa Centre for Flamenco Research, 11004 Cádiz, Spain; vargas@flamencoinvestigacion.es; 4Department of Medicine and Orthopaedic Surgery, Faculty of Medicine, Regional Campus of International Excellence “Campus Mare Nostrum”, University of Murcia, C.P., 30100 Murcia, Spain; santonja@um.es

**Keywords:** back pain, back injuries, risk factors, sagittal spine misalignments, preventive exercise program

## Abstract

The aims of this study were to describe the sagittal integral morphotype (SIM) of classical ballet (CB) dancers, and to establish predictor factors and their cut off values for high risk of experiencing sciatica or low back pain (LBP). This retrospective cohort study was performed in 33 female professional CB dancers. Data related to anthropometric parameters, CB dance experience, sciatica or LBP history, and sagittal spine curvatures were collected. A binary logistic regression and receiver-operating characteristic analysis were performed. The main spine misalignments observed in the SIM of CB dancers were thoracic functional hyperkyphosis, hypomobile kyphosis, and hypokyphosis, and those for the lumbar curvature were hyperlordotic attitude and functional hyperkyphosis. The lumbar curvature in slump sitting and trunk forward bending positions, together with the stature, were significant predictor factors of sciatica history, while the years of dance experience was a significant predictor factor of LBP history. The cut off values analysis revealed that dancers with a stature of 161 cm or less, and those with 14 years of experience or more, have a greater probability of experiencing sciatica or LBP history, respectively.

## 1. Introduction

Classical ballet (CB) is a combination of athletic and artistic activities, where dancers perform complex movement patterns, highly demanding in strength, flexibility, and neuromotor control [1,2]. In addition, this discipline implies particularly high physical and technical demands on the trunk [3]. Repetitive movements of the trunk in non-physiological positions produce very high loads and strains on muscles and joint tissues of the spine, which may finally cause back pain [2,4]. 

The CB dancers show between 0.6 to 4.4 injuries per 1000 h of dance practice [1,2,5,6]. Different studies have investigated back injuries in dancers. The incidence of spine injuries (thoracic and lumbar facet joint dysfunction and nerve root pathology) has been established in a range from 0.12 to 0.34 per 1000 h of dance practice [5]. Gamboa et al., [1] observed back pain in the 9.4% of dancers of a boarding dance school, and nearly one third of the CB and flamenco dancers showed at least one degenerated disc in the study performed by Capel et al. [7]. In addition, the mean spine injury severity has been found in a range of 2 to 9 days loss per 1000 h of dance in professional CB dancers [5]. 

It has been reported that CB dancers show modifications of the normal sagittal curvatures of the spine, which may cause back injury. These changes have been correlated to different factors, such as age (over 18 years old [1,8]), years of training [2], hours of training per week [9], and high spine compressive forces [9]. In this sense, three investigations found a high prevalence of decreased thoracic kyphosis and lumbar lordosis in ballet students [1,9] and experienced CB dancers [10,11] at standing position. In the same body position, Ambegaonkar et al. [12] reported moderate or marked lumbar hyperlordosis in 50% of CB dancers evaluated in their study. 

The assessment of the spine curvatures has been the subject of several studies, because the important association found between sagittal spine misalignments and back pain [13,14,15]. These misalignments have been identified in descriptive studies comparing the spine morphotype of dancers in standing position with the normal values proposed by the authors [1,12] or by comparing with a control group [9]. Ambegaonkar et al. [12] measured the lumbar lordosis by means of a two-dimensional system based on sagittal plane pictures and the Watson MacDonncha Posture analysis instrument. These authors established three categories of lordosis (marked deviation, moderate lumbar lordosis, and no deviation or normal lumbar lordosis) depending on the measurements of the circumference representing the lumbar curvature. Gamboa et al. [1] used the 3-categories classification (flat, normal, or increased) described by Kendall et al. [16] to classify the spine misalignments and rate of injury in elite adolescent ballet dancers. 

However, the studies mentioned above did not evaluate and interpret spine curvatures according to the classification system of the sagittal integral morphotype (SIM). In this sense, Santonja-Medina et al. [17] have published recently the reference values for the spine morphotype according to the classification system of the SIM. These authors combine measurements of the sagittal spine morphotype in the most common positions (standing, slump sitting, and trunk bending forward) that are adopted daily, in fitness and in sports activities. The combination of these three spine morphotypes in the SIM allows an accurate detection of the sagittal spine misalignments, which facilitates the diagnosis of spine injuries [17]. Despite these advantages, this method has not yet been applied in dancers. 

On the other hand, no predictor factors that would help to predict sciatica and back pain in CB dancers have yet been identified [5,18]. In this regard, it would be also very useful to determine cut off values for these predictor factors that may identify an increased likelihood of suffering for both pathologies.

Therefore, the present study aimed to describe the SIM of CB dancers and to identify predictor factors of sciatica and LBP and their cut off values. We hypothesize that there are specific adaptations of the spine to the physical and technical requirements of CB dance that modifies the sagittal spine morphotype of dancers with respect to the control population. In addition, we hypothesize that years of training, anthropometric parameters, and lumbar spine misalignments could be risk factors for sciatica and LBP history in dancers.

## 2. Materials and Methods

### 2.1. Research Design

This retrospective cohort study was performed in 33 female professional CB dancers. This investigation considered demographic and anthropometric data, years of dance training, years of CB training, and sagittal spine curvatures as potential predictive factors for sciatica and LBP in CB dancers. The assessment session was conducted in the academic year 2019/2020. Both, the completion of the questionnaire (years of training dance, years of CB training, history of sciatica and LBP) and the assessment of sagittal spine curvatures (thoracic and lumbar curvatures) were carried out during two days. One session of familiarization was completed three days before the evaluation session. The CB dancers practiced the procedure of each test in this familiarization session. Two experienced evaluators participated in this study. The main evaluator assessed sagittal spine curvatures. The assistant evaluator helped dancers to complete the questionnaire and recorded the anthropometric parameters. The SIM was obtained from the measurement of the thoracic and lumbar curvatures in relaxed standing (RS), slump sitting (SS), and trunk forward bending (TFB) positions. All the CB dancers were asymptomatic at the assessment session. Before performing this study, a preliminary double-blind sagittal spine curvatures study was conducted (2 assessment sessions, 24 h apart) with 12 dancers in order to establish the intra evaluator reliability and intraclass correlation coefficients (ICC). These coefficients were greater than 0.91 (thoracic curvature, 0.92 to 0.98; lumbar curvature, 0.97 to 0.98) for all variables.

### 2.2. Subjects

The study was conducted with professional CB dancers, who had just completed their studies at a Spanish Official Conservatory of Dance. The demographic and anthropometric parameters of the CB dancers are shown in Table 1. The CB dancers had studied at least 8 years of dance and 4 years of CB dance. During the last four years, the weekly training volume was approximately 20 h of CB dance practice per week.

Dancers with orthopedics disorders affecting the lower extremity or back in the last two weeks, and those CB dancers with structural spine pathologies were excluded from this study. None of the participants reported a spine surgery history.

The CB dancers, or parents or legal tutors, signed an informed consent form after being informed about the study procedure. The study was carried out according to the ethical standards of the Helsinki Declaration 1975 and was approved by the Ethics and Research Committee of the University of Murcia (Spain) for studies involving human subjects (ID: 1702/2017).

### 2.3. Questionnaire

The CB dancers completed a questionnaire about their demographic, anthropometric, dance-related background, and systematic training volume data. The information in the questionnaires was cross-referred with the teacher of Official Conservatory of Dance and parents or legal tutors to avoid subjectivity. The assistant evaluator assessed the anthropometric parameters. Dancers were asked if they had experienced sciatica or LBP for longer than one week or whether they did not attend at least three days of dance classes due to sciatica or LBP within the last 12 months [19,20], both sciatica and LBP not being related to trauma or menstrual pain [21].

### 2.4. Evaluation of Potential Predictors Factors for Sciatica and LBP

Anthropometric data (body mass, stature, and body mass index) were measured using a mobile stadiometer (Seca 213; Seca Ltd., Hamburg, Germany). A correction of 0.5 kg was made for the clothes weight. The body mass index (BMI) was calculated as a student’s body mass in kilograms divided by the square of the student’s stature in meters (Kg/m^2^).

Sagittal spine morphotype (thoracic and lumbar curvatures) were examined in the three most common positions (RS, SS, and TFB) following the methodology described by Santonja-Medina [17] (Figure 1). The SIM was determined by combining the three positions. Sagittal spine curvatures were assessed using an ISOMED Unilevel inclinometer (ISOMED, Inc, Portland, OR, USA). Negative values in standing position represented degrees of posterior concavity (lordosis), while positive values indicated anterior concavity or kyphosis.

All the CB dancers were asymptomatic at the evaluation session. Dancers were instructed not to participate in any intensive dance training or intense physical activity 24 h before their assessment. The CB dancers did not perform warm-up or stretching exercises before or during the measurements. The CB dancers were examined in sports bra and short sports leggings to facilitate the positioning of the measuring instrument on the back and sacrum relief. All the measurements were performed the same day. In a blinded controlled study, the anthropometry, and the sagittal spine curvatures were taken in a temporal random order for each CB dancers. Three trials for each measure were performed and the average score of the two nearest measures was employed for data analysis [17].

### 2.5. Statistical Analyses

The statistical data analysis was performed with the SPSS version 24 software (SPSS Inc., Chicago, IL, USA). Firstly, the normal distribution of raw data sets was checked using the Shapiro-Wilk’s test. The level of significance (α) was set at 0.05; therefore, *p* values less than 0.05 were considered statistically significant. 

Descriptive statistics including mean values and standard deviations, as well as absolute and relative frequencies were calculated for each classification and subclassification according to their SIM [17]. Normal and sagittal spine misalignments values in these study were those defined by Santonja-Medina after conducting a clinical-radiological study of the sagittal spine morphotype [22,23]. 

A backward stepwise binary logistic regression was performed to identify predictors factors (demographic and anthropometric characteristics, dance experience, classical dance experience, and sagittal spine curvatures) of sciatica and LBP history (forward selection (enter regression or stepwise regression), inclusion probability *p* ≤ 0.05, elimination probability *p* ≤ 0.10). The resulting odds ratios (OR) and associated 95% confidence intervals (CI) were displayed.

A binary classification analysis by receiver operating characteristic (ROC) was calculated to decide an optimal cut off value for each of the predictor factors analyzed. This statistical analysis identified the CB dancers with high risk of experiencing sciatica and LBP. The highest sensitivity (ability to detect the CB dancers with high risk of sciatica or LBP) and specificity (ability to detect the asymptomatic CB dancers) were determined at the optimal cut off score estimated by the Youden index test. These optimal cut off scores provide the best discriminating capacity between asymptomatic CB dancers and those with sciatica or LBP history. The correlation between the predictor factors (low versus high risk for optimal cut off value) and sciatica or LBP history was obtained by Pearson’s chi-squared statistic.

## 3. Results

According to the questionnaire, 11 CB dancers have experienced sciatica and 16 from LBP in the 12 months preceding the study. 

Firstly, thoracic and lumbar curvatures were assessed individually in each position (RS, SS, and TFB). The results related to the absolute and relative frequencies for each parameter according to the normal values [17] are represented in Table 2.

Subsequently, the thoracic SIM was obtained according to the combined spinal morphotype related to the three positions evaluated in the CB dancers. Table 3 displays the absolute and relative frequencies of each category of the thoracic SIM and the results recorded in each position. In total, 17 CB dancers presented normal morphotype (51.5%) since their values were normal in the three measurement positions, 9 CB dancers were classified as functional thoracic hyperkyphosis (27.3%), four CB dancers as hypomobile kyphosis (12.1%), and three CB dancers as thoracic hypokyphosis or hypokyphosis attitude (9.1%).

Similarly, the lumbar SIM classification of the CB dancers is represented in Table 4. Overall, 23 CB dancers displayed a normal morphotype (69.7%) or normal lumbar curvature in all positions, six CB dancers were classified as lumbar hyperlordotic attitude (18.2%), and four were classified as functional lumbar hyperkyphosis (18.2%).

The first stepwise logistic and enter regression analysis of the potential predictor factors for sciatica history (Table 5) showed that lumbar curvature in SS (sensibility = 81.2% vs specificity = 88.2%), lumbar curvature in TFB (sensibility = 81.2% vs specificity = 88.2%) and stature (sensibility = 68.7% vs specificity = 82.3%) had high classification accuracy for CB dancers with sciatica history (28 of 33 students for lumbar curvature (84.8%) and 25 of 33 students for stature (75.8%)). This statistical analysis indicated that years of dance experience had a high classification accuracy (25 of 33 students (75.8%)) for CB dancers with or without an LBP history (sensibility = 45.4% vs specificity = 90.9%). Stepwise logistic regression analysis showed that among the potential predictor factors for sciatica (Table 5) entered into the model, the lumbar curvature in SS (OR = 1.420, 95% CI = 0.500 to 0.989, *p* = 0.043), lumbar curvature in TFB (OR = 1.623, 95% CI = 1.003 to 2.626, *p* = 0.048) and stature (OR = 1.232, 95% CI = 0.664 to 0.992, *p* = 0.042) were medium predictor factors of sciatica history. The year of dance experience was a medium predictor factor (OR = 1.250, 95% CI = 1.012 to 1.543, *p* = 0.038) of LBP history (Table 5).

The area under the ROC curve for years of dance experience and stature was 0.740 and 0.721, respectively (Figure 2), being statistically significant (years of dance experience (*p* = 0.027, standard error: 0.095, 95% confidence interval: 0.553 to 0.926) and stature (*p* = 0.031, standard error: 0.095, 95% confidence interval: 0.093 to 0.466)). The cut off values for years of dance experience and stature that identified most accurately dancers with high risk for LBP and sciatica were 14 years (sensibility 0.727 vs 0.273 specificity) and 161 cm (sensibility 0.882 vs 0.500 specificity), respectively.

The chi-square test revealed that having a small stature (≤161 cm) was associated with having sciatica in the previous year (X2_(33)_ = 7.340; *p* = 0.007; η^2^ = 0.472). The estimation of the relative risk showed that CB dancers with lower stature (≤161 cm) had 9.6 times higher risk of developing sciatica [95% CI = 1.633 − 56.925] than CB dancers with a superior stature (>161 cm). This statistical analysis also determined that having a large training experience (≥ 14 years) was associated with having LBP in the previous year (X2_(33)_ = 4.739, *p* = 0.029, η^2^ = 0.397). Those dancers with more or equal than 14 years of training experience had 5.8 times higher risk of developing LBP [95% CI = 1.118 − 29.847] than the less experienced dancers.

## 4. Discussion

This is the first report describing the SIM in CB dancers and it is also the first study identifying predictors factors for sciatica and LBP in this population. Dancers showed a higher percentage of thoracic hyperkyphosis in SS (18.2%) and TFB (9.1%) than that in RS (0%). For thoracic hypokyphosis, similar results were observed, with a higher frequency of this spine misalignments in SS (15.2%) and TFB (18.2%) compared to those in RS (6.1%). With regard to the spine misalignments of the lumbar curvature, the highest percentage was observed in RS (hyperlordosis: 18.2%) compared to those shown in SS (hyperkyphosis: 12.1%) and TFB (hyperkyphosis: 6.1%).

The combined assessment of the spine misalignments in the three positions contributes to a more accurate diagnosis of possible spine pathologies [13,17,22]. In this sense, the analysis of the SIM of CB dancers showed mainly four of the 13 categories of misalignments defined previously by Santonja-Medina for the thoracic curvature [17], which were functional hyperkyphosis (27.3%), hypomobile kyphosis (12.1%), and hypokyphosis or hypokyphosis attitude (9.1%). An increased thoracic curvature is caused by weakness of the trunk erector and scapular retraction muscles (trapezius, rhomboids, and latissimus dorsi) in SS (functional hyperkyphosis) and in TFB (functional hyperkyphosis), together with pectoralis minor tightness [25]. On the contrary, several authors [9,11] have suggested that the repetition of specific and sustained movements of the CB dance such as the *Port de* Bras Devant, Circle Port de Bras, or Cambré Derrière cause the decrease in thoracic curvature in RS (hypokyphosis) and TFB (hypokyphosis and hypomobile kyphosis).

For the lumbar curvature only two out of the 17 categories of the SIM misalignments defined by por Santonja-Medina [17] were observed; hyperlordotic attitude (18.2%), and functional lumbar hyperkyphosis (12.2%). Increased values of the lumbar curvature in RS (hyperlordotic attitude), SS (functional lumbar and hyperkyphosis static and total), and TFB (hyperkyphosis static total) are caused by tightness of the hip flexors, and weakness of the trunk flexors and gluteus maximus [26,27,28]. The increased lumbar lordosis in the RS (hyperlordotic attitude) is a consequence of the lumbar spine pattern acquired by the repetition of technical movements of the CB dance such as En Dehors, Cambré Derrière, or Grand Battement [11]. In this regard, the hyperkyphosis in SS and TFB may be a consequence of the repetition of the Port de bras devant movement. Furthermore, we should bear in mind that the inadequate postures of the CB dancers in their day-to-day life may also have an impact on the definition of the spine misalignments found in this study.

The SIM analysis performed in this study has been previously made in athletes of different sports. The percentages of sagittal spine misalignments in the present study are much lower than those observed in equestrian athletes [29], inline hockey players [13], and artistic gymnasts [30]. The posture training, which is a fundamental aspect of the dancers’ technical gesture, may explain these differences.

This study demonstrates that three of the variables related to demographic aspects, sport, anthropometric parameters, and sagittal spine disposition were predictors factors for sciatica and LBP in CB dancers. The regression model found three predictors (lumbar curvature in SS, lumbar curvature in TFB and stature) for history of sciatica (*p* ≤ 0.048) and one predictor (years of dance experience) for LBP in CB dancers (*p* = 0.038). These expected results are the main findings of the present study and will be highly useful to reduce the likelihood of experiencing sciatica and LBP in CB dancers.

In the present investigation cut off values at 161 cm for stature, 8° for lumbar curvature in SS and 24.5° for lumbar curvature in TFB were established. The CB dancers with values equal to or less (stature)/more (lumbar curvature in SS and TFB) than these cut off values have an increased risk for developing sciatica. Considering these cut off values, the likelihood of experiencing sciatica was 72.1% for stature, 45.6% lumbar curvature in SS, and 57% for lumbar curvature in TFB. The short (≤161 cm) CB dancers are challenged to perform the technical movements with greater ROM to match the stage presence of the tall dancers. The repetition of external rotation movements and hip abduction, which are typical of the basic joint movements of CB, predisposes to piriformis syndrome [31]. Both repetitive hip movements, stabilization of the pelvic girdle, and weakness hip flexors and abductors may cause piriformis syndrome [31,32]. These demands require the high and constant activation of the piriformis muscle which compresses the sciatic nerve [32]. The execution of these movements (external rotation and hip abduction), with a higher demand by CB dancers with low stature, reasonably leads to a faster onset of piriformis syndrome [31,32]. Piriformis syndrome leads to pathologic conditions of the sciatic nerve, chronic somatic dysfunction, and compensatory changes resulting in pain, paresthesia, hypesthesia, and muscle weakness [31,33]. This argument is supported by a systematic review of Swain, Bradshaw, Whyte, and Ekegren [18], who found an association between stature and LBP in pre-professional and professional dancers. On the contrary, the stature has also been associated with the LBP in military helicopter pilots [34], adult population aged from 30 to 69 years [35], and community-dwelling elderlies [36].

Considering the LBP, an optimal cut off was determined for years of dance experience. The CB dancers with an experience equal to or higher than 14 years have a 74% of probability of suffering LBP. Previous studies have not considered the years of experience as a risk factor for LBP, but they have considered the age [8,18]. Most investigations about dance injuries have been performed on mature (age > 18 years) female professional dancers (mainly CB dancers), assuming that at these ages the dancers have many years of experience [1]. The LBP in CB dancers is due to acumulative effects over the years of a suboptimal control motor such as poor postural alignment, lack of coordination or incorrect technique executions [1,37,38], together with muscle imbalances [1,2], hypermobility or muscle tightness [1,2,37], the maximal turnout in the lower limbs [39], and the excessive repetitive movements in non-physiological positions [7,9,32].

The LBP is also caused by the lumbar hyperlordosis observed in some movements such as the Backbends and Arabesques [40]. Pelvic flexion and retroversion movements performed in Backbends and Arabesques have been found to increase the risk of LBP [18,37]. These movements produce lumbar hyperlordosis and anteriorly pelvic tilting in dancers with weak abdominal muscles and tight thoracolumbar fascia [40]. Moreover, Backbends and Arabesques are highly demanding movements of external hip rotation [32,40,41], which may also cause hyperlordosis and pelvic retroversion in case of musculoskeletal imbalance in flexibility and strength [1,37]. All these factors create a very high load and strain on muscles, intervertebral disc, and ligaments, which predispose dancers to suffer spondylolysis, spondylolisthesis, and lumbar facet sprain. All these injuries manifest themselves with LBP [32]. This situation, in which the physical and technical demands of the dance involve hyperlordosis and anteversion of the pelvis, has been described in dancers [2,18,42], and in athletes of different sports, such as artistic gymnastics [30], rhythmic gymnastics [43], trampoline gymnastics [44,45] or figure skating [46].

### Practical Recommendations

Based on the results of the present study, a training program for strength, flexibility, and posture is recommended for CB. This program must be individualized for the short dancers, because they must execute the technical movements of CB with a higher range of movement than their tall counterparts without excessive stress on the joint tissues. In addition, this training program should control the risk factors associated with the piriformis syndrome found in the present study, which causes sciatica [31,33] and LBP [47]. Specifically, this training regimen should include stretching exercises for the hip rotator, abductor and adductor muscles (particularly the trigger points in the piriformis muscle), strengthening of adductors and abductors, a stabilization in an appropriate pelvis arrangement in the flexion and extension movements of the trunk and alignment of sacroiliac joint [31,33,48]. These recommendations should be applied from the beginning of the dance practice considering that the years of practice is a significant risk factor for LBP. The training program should be accompanied by a periodical assessment of the SIM, in order to avoid spine pathologies associated to the dance practice.

## 5. Conclusions

The main spine misalignments observed in the CB dancers were thoracic functional hyperkyphosis, hypomobile kyphosis, and hypokyphosis, and those for the lumbar curvature were hyperlordotic attitude and functional hyperkyphosis.

The lumbar curvature in SS and in TFB, together with the stature were predictor factors of sciatica history. Number of years of CB experience was a predictor factor of LBP history in the dancers. Dancers with a stature of 161 cm or less, and dancers with 14 years of experience or more had a greater probability of experiencing sciatica or LBP, respectively.

## Figures and Tables

**Figure 1 ijerph-18-05039-f001:**
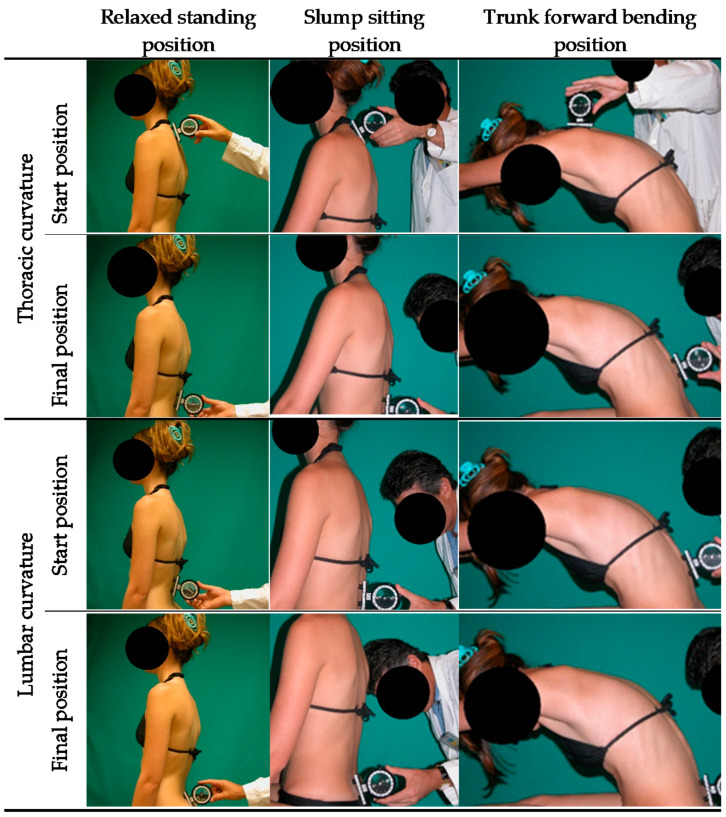
Tests for the assessment of the sagittal integral morphotype [17].

**Figure 2 ijerph-18-05039-f002:**
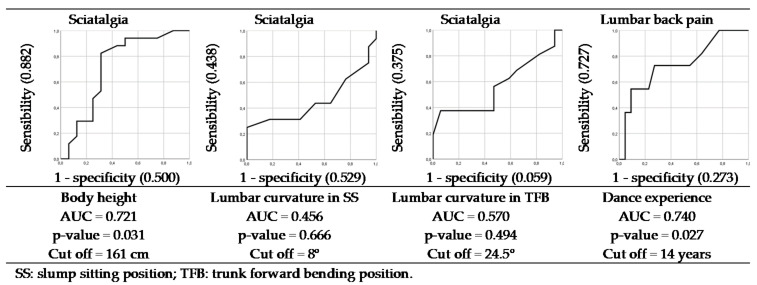
Sensibility, specificity, and area under curve for predictor factors statistically correlated to sciatica and lumbar back pain in classical ballet dancers.

**Table 1 ijerph-18-05039-t001:** Demographic and anthropometric parameters of the 33 classical ballet dancers.

Variables	Minimum Value	Maximum Value	Total Value ^1^
Age (years)	16.0	31.0	22.0 ± 3.8
Body mass (kg)	43.5	59.6	51.9 ± 4.4
Stature (cm)	150.0	170.5	161.7 ± 4.5
Body mass index (kg/m^2^)	16.3	23.1	19.9 ± 1.7
Dance experience (years)	8.0	23.0	13.4 ± 4.0
Classical ballet dance experience (years)	4.0	23.0	8.8 ± 5.4

^1^ Values are expressed as mean ± standard deviation.

**Table 2 ijerph-18-05039-t002:** Absolute and relative frequencies of the spine curve in the three positions assessed according to normal values in female classical ballet dancers.

Variable	Position	Category ^1^	Total Value	N ^2^	% ^3^
Thoracic curvature	RS	Hypokyphosis (<20°)	14.0 ± 5.7	2	6.1
		Normal (20 to 40°)	29.2 ± 6.8	31	93.9
		Hyperkyphosis (≥41°)	−	−	−
	SS	Hypokyphosis (<20°)	14.4 ± 2.9	5	15.2
		Normal (20 to 40°)	32.8 ± 5.9	22	66.7
		Hyperkyphosis (≥41°)	49.8 ± 6.7	6	18.2
	TFB	Hypokyphosis (<40°)	32.8 ± 4.0	6	18.2
		Normal (40° to 65°)	51.5 ± 7.3	24	72.7
		Hyperkyphosis (≥66°)	69.0 ± 1.7	3	9.1
Lumbar curvature	RS	Hypolordosis (<−20°)	−	−	−
		Normal (−20° to −40°)	−33.1 ± −5.0	27	81.8
		Hyperlordosis (>−40°)	−44.3 ± 0.8	6	18.2
	SS	Hypokyphosis (<−15°)	−	−	−
		Normal (−15° to 15°)	6.8 ± 5.3	29	87.9
		Hyperkyphosis (>15°)	19.3 ± 1.5	4	12.1
	TFB	Hypokyphosis (<10°)	−	−	−
		Normal (10° to 30°)	19.0 ± 5.1	31	93.9
		Hyperkyphosis (>30°)	32.0 ± 0.0	2	6.1

Data are expressed as Mean ± SD; RS: relaxed standing position; SS: slump sitting position; TFB: trunk flexion bending position; LSA: lumbosacral angle; ^1^ Normal and sagittal spine misalignments values were defined according to Santonja-Medina et al. [17]; ^2^ Number of cases; ^3^ Number of cases with respect to the total of female classical ballet dancers evaluated.

**Table 3 ijerph-18-05039-t003:** Absolute and relative frequencies of each category of the thoracic curvature with respect to the sagittal integral morphotype (SIM).

Thoracic SIM ^1^		Position			N ^2^	% ^3^
Category	Subcategory	RS	SS	TFB		
Normal Kyphosis		Normal (20° to 40°)	Normal (20° to 40°)	Normal (40° to 65°)	17	51.5
Hypokyphosis or hypokyphosis attitude	Dynamic	Hypokyphosis (<20°)	Normal (20° to 40°)	Hypokyphosis (<40°)	3	9.1
Hypomobile kyphosis		Normal (20° to 40°)	Normal (20° to 40°)	Hypokyphosis (<40°)	4	12.1
Functional hyperkyphosis	Static	Normal (20° to 40°)	Hyperkyphosis (>40°)	Normal (40° to 65°)	6	18.2
	Dynamic	Normal (20° to 40°)	Normal (20° to 40°)	Hyperkyphosis (>65°)	2	6.1
	Total	Normal (20° to 40°)	Hyperkyphosis (>40°)	Hyperkyphosis (>65°)	1	3

Data are expressed as Mean ± SD; RS: relaxed standing position; SS: slump sitting position; TFB: trunk flexion bending position.^1^ Classification of thoracic integrative morphotype was defined according to Santonja-Medina et al. [17]; ^2^ Number of cases; ^3^ Number of cases with respect to the total of female classical ballet dancers evaluated.

**Table 4 ijerph-18-05039-t004:** Absolute and relative frequencies of each category of the lumbar curvature with respect to the sagittal integral morphotype (SIM).

Lumbar SIM ^1^		Position			N ^2^	% ^3^
Category	Subcategory	RS	SS	TFB		
Normal lumbar curve		Normal (−20° to −40°)	Normal (−15° to 15°)	Normal (10° to 30°)	23	69.7
Hyperlordotic attitude		Hyperlordosis (>−40° )	Normal (−15° to −15° )	Normal (10° to 30° )	6	18.2
Functional lumbar hyperkyphosis	Static	Normal (−20° to −40°)	Hyperkyphosis (>15°)	Normal (10° to 30°)	2	6.1
	Total	Normal (−20° to −40°)	Hyperkyphosis (>15°)	Hyperkyphosis (>30°)	2	6.1

Data are expressed as Mean ± SD; RS: relaxed standing position; SS: slump sitting position; TFB: trunk flexion bending position.^1^ Classification of lumbar integral morphotype was defined according to Santonja-Medina et al. [17]; ^2^ Number of cases; ^3^ Number of cases with respect to the total of female classical ballet dancers evaluated.

**Table 5 ijerph-18-05039-t005:** Absolute frequencies and logistic regression results for predictive factors of sciatica and low back pain (LBP).

Method Sample	Risk Factors		OR ^1^	SE	95% CI	*p*-Value
Enter Regression	LC in SS		1.420 Medium	0.174	0.500 to 0.989	0.043
	≥8° ^1^	<8°				
Asymptomatic (n = 17)	52.9%	47.1%				
Sciatica (n = 16)	62.5%	37.5%				
Enter Regression	LC in TFB		1.623 Medium	0.245	1.003 to 2.626	0.048
	≥24.5° ^1^	<24.5°				
Asymptomatic (n = 17)	5.9%	94.1%				
Sciatica (n = 16)	62.5%	37.5%				
Stepwise Regression	Stature					
	≤161 cm ^1^	>161 cm	1.232 Small	0.102	0.664 to 0.992	0.042
Asymptomatic (n = 17)	64.7%	35.3%				
Sciatica (n = 16)	75.0%	25.0%				
Stepwise Regression	Dance experience		1.250 Medium			0.038
	≥14 y ^1^	<14 y		0.107	1.012 to 1.543	
Asymptomatic (n = 22)	31.6%	68.4%				
Low back pain (n = 11)	72.7%	27.3%				

LC: lumbar curvature; SS: slump sitting position; TFB: trunk forward bending position; SE: Standard Error; CI: Confidence Interval. ^1^ OR: Odds Ratio (relative risk); OR < 1: poor predictor of LBP; OR from 1 to 1.25: small predictor; OR from 1.25 to 2: medium predictor; OR ≥ 2: large predictor [19,24].

## Data Availability

The data sets used and analyzed during the current study are available from the first or corresponding author on reasonable request.

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
