# Peer review of "Sagittal Integral Morphotype of Female Classical Ballet Dancers and Predictors of Sciatica and Low Back Pain"

_ijerph, 2021, doi:10.3390/ijerph18095039_

Round 1

Reviewer 1 Report

Well done on the study. The main issue of the study is related to the typos/grammar/etc. I have listed the ones I have identified, but they were so prevalent that it became frustrating to read the paper at a deeper level. 

Page 1, Line 18 – typo – establish

Page 2 – Line 62 – typo. Establish

Page 2 – Line 86 – grammatically incorrect – predictors variables of sciatica

Page 4, line 151 – grammatically doesn’t make sense

Page 5, Line 192 The 100% of the CB dancers showed - grammar

Page 5 – Table 2.  You have a category as ‘rectification’. I am not sure what this category means. Rectification is the action of putting something right. I’m not sure hypo-lordosis or hypo-kyphosis is rectification. Reviewing Santonja-Medina et al (2020), I can see that rectification was used in their results table 4, but in their Tables 2 and 3, rectification is not used as a classification term. Is this an oversight of the 2020 paper and not picked up or is there a reason for using rectification.

Page 4-5, Lines 188-196 – This paragraph is just the worded version of Table 2 – Seems to be redundant.

Page 6 – a significant number of typos – sensibility vs sensitivity

Page 8 :Line 252 grammar

Page 8 Line 265 - shoulder retropulsor muscles?

Author Response

Dear Reviewer,

Thank you for your review and constructive comments. We have reviewed the manuscript and addressed all your remarks (see the responses below). We hope that you consider the reviewed version of the manuscript worthy for publication.

Kind regards,

The authors.

Comments and Suggestions for Authors

Well done on the study. The main issue of the study is related to the typos/grammar/etc. I have listed the ones I have identified, but they were so prevalent that it became frustrating to read the paper at a deeper level. 

Response: Professor Mark De Ste Croix (native English) from the University of Gloucestershire and Dr Cristina Cuello from the University of Murcia have carried out an extensive revision of the manuscript and we have corrected the grammatical errors detected.

Page 1, Line 18 – typo – establish

Response: Done.

Page 2 – Line 62 – typo. Establish

Response: Done.

Page 2 – Line 86 – grammatically incorrect – predictors variables of sciatica

Response: Thank you for the comment. We have rewritten this sentence.

Page 4, line 151 – grammatically doesn’t make sense

Response: Thank you for the comment. We have rewritten this sentence.

Page 5, Line 192 The 100% of the CB dancers showed - grammar

Response: Thank you for the comment. We have deleted this paragraph following the recommendations of a comment below.

Page 5 – Table 2.  You have a category as ‘rectification’. I am not sure what this category means. Rectification is the action of putting something right. I’m not sure hypo-lordosis or hypo-kyphosis is rectification. Reviewing Santonja-Medina et al (2020), I can see that rectification was used in their results table 4, but in their Tables 2 and 3, rectification is not used as a classification term. Is this an oversight of the 2020 paper and not picked up or is there a reason for using rectification.

Response: Correct. Rectification is the correction or reduction of the physiological curvatures of the spine in the sagittal plane. In order to avoid confusing terminology in the paper, rectification” has been replaced by “hypo-lordosis” or “hypo-kyphosis” throughout the manuscript.

Page 4-5, Lines 188-196 – This paragraph is just the worded version of Table 2 – Seems to be redundant.

Response: The paragraph has been deleted.

Page 6 – a significant number of typos – sensibility vs sensitivity

Response: Corrected.

Page 8: Line 252 grammar

Response: Revised and corrected.

Page 8 Line 265 - shoulder retropulsor muscles?

Response: shoulder retropulsor muscles has been replaced by scapular retraction muscles.

Reviewer 2 Report

Dear Authors, please find my remarks below:

l.18 establish instead of stablish

l. 29 I do not understand: „greater odd of suffering”

l. 90 type of study should be clarified (retrospective cohort?)

l 128 We do not have an anthropometric variable (what should change?) but parameters/data

tab. 1 is not a result, this should be in the subject subsection

tab. 1 BMI index is useless, especially for people performing sport, it should be replaced by body composition or body fat amount

l. 342, In general, dancers present lower levels of fitness as compared to other athletes and the healthy general population of similar age [43]. This statement is inconsistent with my experience and the data that I know. Generally, professional dancers have higher stamina and are fit, but at the same time often suffer from degeneration of the locomotor system and are more likely to be injured. In classical ballet, the most problematic is the level of the same movement repetitions and force impulses appearing during impacts (ex. foot contact with the ground) and artificially increased range of motion. This part needs a further explanation and correction

In the work, there is an unacceptable level of self-citation

I think that the language should be corrected by the specialist/native speaker. 

Results section - it is difficult to follow the authors' idea of presentation - needs revision.

Author Response

Dear Reviewer,

Thank you for your review and constructive comments. We have reviewed the manuscript and addressed all your remarks (see the responses below). We hope that you consider the reviewed version of the manuscript worthy for publication.

Kind regards,

The authors.

Comments and Suggestions for Authors

Dear Authors, please find my remarks below:

l.18 establish instead of stablish

Response: Done.

  1. 29 I do not understand: „greater odd of suffering”

Response: We have rewritten this sentence.

  1. 90 type of study should be clarified (retrospective cohort?)

Response: Done.

l 128 We do not have an anthropometric variable (what should change?) but parameters/data

Response: Corrected.

tab. 1 is not a result, this should be in the subject subsection

Response: Corrected.

tab. 1 BMI index is useless, especially for people performing sport, it should be replaced by body composition or body fat amount

Response: Thank you for the comment. We will measure body composition or body fat amount in future studies.

  1. 342, In general, dancers present lower levels of fitness as compared to other athletes and the healthy general population of similar age [43]. This statement is inconsistent with my experience and the data that I know. Generally, professional dancers have higher stamina and are fit, but at the same time often suffer from degeneration of the locomotor system and are more likely to be injured. In classical ballet, the most problematic is the level of the same movement repetitions and force impulses appearing during impacts (ex. foot contact with the ground) and artificially increased range of motion. This part needs a further explanation and correction

Response: Since this sentence is controversial, we have decided to delete it. In addition, it does not fit in the Practical recommendations section.

In the work, there is an unacceptable level of self-citation

Response: We have reduced the number of self-citations by keeping the most important ones.

I think that the language should be corrected by the specialist/native speaker. 

Response: Professor Mark De Ste Croix (native English) from the University of Gloucestershire and Dr Cristina Cuello from the University of Murcia have carried out an extensive revision of the manuscript and we have corrected the grammatical errors detected.

Results section - it is difficult to follow the authors' idea of presentation - needs revision.

Response: We have revised this section in order to improve text clarity.

Reviewer 3 Report

Spelling error line 42

The method would be improved with images of the  measurements being undertaken  

A reference is needed to justify why you took the average of 2 measures 

In the discussion the term 'suffering' when you refer to  sciatica might be better with 'experiencing' sciatica

The practical recommendations section which refers to physical fitness seems unrelated to the data you have collected so I suggest remove this and add something that is relevant.

Author Response

Dear Reviewer,

Thank you for your review and constructive comments. We have reviewed the manuscript and addressed all your remarks (see the responses below). We hope that you consider the reviewed version of the manuscript worthy for publication.

Kind regards,

The authors.

Comments and Suggestions for Authors

Spelling error line 42

Response: Corrected.

The method would be improved with images of the measurements being undertaken  

Response: Done.

A reference is needed to justify why you took the average of 2 measures.

Response: Done.

In the discussion the term 'suffering' when you refer to sciatica might be better with 'experiencing' sciatica

Response: Done.

The practical recommendations section which refers to physical fitness seems unrelated to the data you have collected so I suggest remove this and add something that is relevant.

Response: We have revised this section and deleted text not related to the data.